# Review of the Application of Microwave Heating Technology in Asphalt Pavement Self-Healing and De-icing

**DOI:** 10.3390/polym15071696

**Published:** 2023-03-29

**Authors:** Letao Zhang, Zihan Zhang, Weixiao Yu, Yinghao Miao

**Affiliations:** National Center for Materials Service Safety, University of Science and Technology Beijing, Beijing 100083, China

**Keywords:** asphalt pavement, microwave heating, pavement de-icing, self-healing

## Abstract

In the past decades, a large amount of research was conducted to investigate the application prospect of microwave heating technology in improving the efficiency of asphalt pavement self-healing and de-icing. This paper reviewed the achievements in this area. Firstly, the properties of asphalt concrete after microwave heating were summarized, including microwave sensitivity and heating uniformity. Then, the evaluation indicators and influence factors of the self-healing properties of the asphalt mixtures heated by microwave were reviewed. Finally, the application of microwave heating in asphalt pavement de-icing was explored. In addition, asphalt pavement aging due to microwave heating was also reviewed. It was found that microwave heating technology has good prospects in promoting asphalt pavement self-healing and de-icing. There are also some problems that should be studied in depth, such as the cost-effectiveness of microwave-sensitive additives (MSAs), the performance of the pavement with MSAs, mechanism-based self-healing performance indicators, and the aging of asphalt pavements under cycling microwave heating.

## 1. Introduction

Asphalt pavements have the advantages of high smoothness and comfortableness and they are the main form of road pavements. Asphalt mixture is a composite material formed using aggregates, fillers, asphalt binders, and voids. Asphalt binders are often used to bind the aggregates and their viscosity depends on external temperature. They behave as a Newtonian fluid at high temperatures, while their behavior is non-Newtonian at low temperatures. Under the long-term influence of climatic conditions and vehicle loads, asphalt pavements will inevitably be damaged, with cracks, potholes, and ruts appearing. In addition, icy roads caused by freezing rain and snowy weather bring potential safety hazards to traffic. Traditional pavement maintenance and de-icing methods have the disadvantages of inefficiency and serious environmental pollution. Researchers have been developing new technologies of pavement maintenance and de-icing, such as induction heating, Joule heating, and microwave heating.

The microwave is actually an electromagnetic wave with a frequency of 300 MHz to 300 GHz and a wavelength of 1 mm to 1 m. Energy can be transferred through space or medium in the form of electromagnetic waves [1]. Microwave heating is a process in which electromagnetic energy is converted into heat. Microwave heating technology possesses many advantages, such as high efficiency, low pollution, energy conservation, and uniform heating, as well as easy control. It has been widely used in cooking, drying, pasteurizing, and other fields [2]. The research in the field of pavement engineering using microwave heating can be traced back to 1974. Bosisio et al. [3] found that microwave radiation with 2450 MHz can effectively penetrate the asphalt concrete layer up to 12 cm. They also used microwave equipment with 1.6 kW to heat asphalt concrete slabs and concluded that actual asphalt pavement cracking can be well repaired using microwave heating. In 1989, Osborne et al. [4] found that the ice layer is facilely separated from the pavement surface under microwave action. Microwave heating has attracted researchers’ attention for its potential in asphalt pavement maintenance and de-icing.

At present, the study on asphalt pavements using microwave heating mainly involves self-healing and de-icing. The principle of self-healing is that asphalt binders can flow and the cracks can be filled when the temperature of asphalt pavements is high. The principle of de-icing is that the microwave can directly heat asphalt concrete pavements through the ice layer and weaken the bond between the ice and the pavement surface. The influence factors of self-healing and de-icing include the properties of asphalt binders and aggregates, temperature, and heating modes. The qualities of self-healing and de-icing mainly depend on the microwave absorption ability of pavement materials. For enhancing the efficiency and effect of asphalt pavement self-healing and de-icing, researchers have carried out investigations from the aspects of microwave-sensitive additives (MSAs) and microwave heating devices [5,6,7]. There have also been a few studies on the aging of asphalt binders and mixtures heated by the microwave and the influence of MSA on pavement performance.

This paper reviewed the investigation into asphalt pavement self-healing and de-icing using microwave heating. The influences of different MSAs and heating conditions on the efficiency of microwave heating, the homogeneity of heating, and the self-healing performance of asphalt concrete were comparably analyzed. The achievements in de-icing by microwave heating were summarized. The research on the potential aging of asphalt under microwave heating was also reviewed. Finally, the issues that still need to be improved were proposed accordingly.

## 2. Microwave Heating Properties of the Asphalt Mixtures

The mechanism of microwave heating is the dielectric loss of materials under the microwave field, including polarized relaxation loss and conductive loss. When asphalt mixture is exposed to microwave radiation, heat is generated through the conversion of the energy of the electromagnetic field. In the conventional heating methods, energy is transferred from the surface of the materials by convection, conduction, and radiation. In contrast, microwave heating is achieved by molecular excitation inside the material without relying on the temperature gradient [8]. Figure 1 describes the microwave-absorbing principle of materials.

### 2.1. Microwave Sensitivity

The microwave sensitivity of a material determines its microwave absorption ability and microwave heating efficiency (MHE). It is commonly represented by dielectric properties. The complex permittivity (*ε*) and complex permeability (*μ*) are two indispensable parameters characterizing the performance of materials reflecting and absorbing microwaves, which are defined as Equation (1) [8].
(1)$$\varepsilon =\varepsilon^{\prime }-j\varepsilon^{\prime \prime },\mu =\mu^{\prime }-j\mu^{\prime \prime }$$
where *ε*′ and *μ*′ are the real parts, indicating the extent of polarization or magnetization of microwave-absorbing materials under the action of electric or magnetic fields, respectively, and *ε*″ and *μ*″ are the imaginary parts, representing the loss magnitude caused by the rearrangement of electric- or magnetic-coupling moments of microwave-absorbing materials under the action of external electric or magnetic fields, respectively. The greater *ε*″ and *μ*″, the better the ability of the material to absorb microwaves. Therefore, the microwave heating properties can be improved by increasing the *ε*″ and *μ*″ of the asphalt mixture.

In addition, the matching attenuation features of materials are also two important conditions for achieving efficient microwave absorption. The former refers to the proportion of the incident microwaves entering materials, which depends on the input wave impedance of the interface between materials and free space. Only when materials match the wave impedance in free space, can the incident microwave get into the material to a large extent. The attenuation characteristics of materials refer to the rapid absorption and attenuation capacity of the microwave entering the materials, which requires that the materials have large electromagnetic loss. This depends on the physical performance of the material itself. The reflection loss (RL) represents the amount of microwave energy reflected by materials. It is another indicator used to evaluate the microwave absorption capacity (MAC) of materials, which can be calculated by Equations (2) and (3) [4]. When the reflection loss is negative, the smaller its value, and the stronger the MAC of materials. The reflection loss values of some representative materials used in asphalt mixtures are summarized in Table 1.
(2)$$R_{L}=20\mathrm{lg}\left|\frac{Zn_{in}-Zn_{0}}{Zn_{in}+Zn_{0}}\right|$$
(3)$$Zn_{in}=Zn_{0}\sqrt{\mu_{r}/\varepsilon_{r}}\tan h\left[j\left(\frac{2\pi fd}{c}\right)\sqrt{\mu_{r}\varepsilon_{r}}\right]$$
where $Zn_{in}$ is the input impedance of materials, $Zn_{0}$ is the intrinsic impedance at free space, $\varepsilon_{r}$ and $\mu_{r}$ indicate the relative complex permittivity and permeability of materials, respectively, f is the electromagnetic frequency, d is the thickness of materials, and c is the speed of light in free space.

Asphalt binders and aggregates are the major components of the asphalt mixtures, while their microwave-absorbing ability is poor. For improving the microwave-absorbing efficiency of the asphalt mixture, researchers developed microwave-absorbing and magnetite-bearing aggregates. Other attempts have included the addition of graphite, carbonyl iron powders (CIPs), carbon nanotubes, and steel wool, as well as ferrite particles. Liu et al. [19] added activated carbon powder (ACP) into asphalt mixture for enhancing the MHE. It was found that the addition of ACP significantly improves the heating efficiency of asphalt mastic and mixture. Zhu et al. [20] studied the heating effect of the asphalt mixture with ferrite and concluded that the MHE of Ni-Zn ferrite powders is 3.91 times as much as that of mineral powder (MP). Wang et al. [21] pointed out that magnetite aggregate has better MHE compared with basalt aggregate. Zhang et al. [22] explored the MHE of ceramics prepared from low-grade pyrite cinder. The results showed that the MHE of the ceramics was far higher than that of limestone. Trigos et al. [23] studied the MHE of various aggregates and graded them. The results showed that the MHE of blast furnace slag, classified as having very high susceptibility, is 24 times higher than that of quartzite, classified as having very low susceptibility. Deng et al. [15] and Cao [24] found that the MHE of basalt is 1.46–5.58 times as much as that of limestone. Li et al. [25] conducted a microwave heating test of asphalt mastics and found that as the volume ratio of the slag filler to asphalt binder increases, the average heating rate of asphalt mastics gradually improves. Li et al. [26] synthesized SiC to wrap LDHs (SwL) at different temperatures and studied the effect of different combinations of SBS and SwL on the microwave absorption of modified asphalt. The results showed that the microwave absorption of SBS/SwL-200-modified asphalt is best. The heating rates of some other typical asphalt mixtures are summarized in Table 2.

The research results mentioned above indicate that the higher the oxide content of iron in the aggregate, the faster the heating rate. The MHE of the asphalt mixture can be promoted by the addition of microwave-sensitive powders. However, the experimental methods of microwave heating for aggregates and fillers are not standardized and unified, which means that the results from various sources cannot be compared. In addition, the cost of MSAs is often ignored in the studies.

### 2.2. Heating Uniformity

In the microwave heating process, the differences in the microwave absorption capacity and heat transfer performance of different materials may lead to uneven temperature distribution in asphalt pavements and large internal temperature gradient [32]. In addition, microwave heating relies on the action of the high-frequency alternating electromagnetic field to realize the transformation of electric energy and material heat energy. In the actual heating process, the electromagnetic field strength is usually distributed unevenly, which can also result in uneven temperature distribution [33].

Many studies were carried out from the aspects of microwave heating modes, microwave heating devices, and MSAs to improve heating uniformity. Zhu et al. [34] simulated intermittent and continuous microwave heating and found that intermittent microwave heating could produce more uniform temperature distribution compared with continuous microwave heating. Sun et al. [35] found that a 0.5 min intermission can bring about a more uniform temperature distribution than no intermission, 1 min intermission, and 2 min intermission. Sun et al. [36] investigated the energy distribution of the electromagnetic field in asphalt mixtures based on the Poynting theory and established an optimized model of the electromagnetic field and structure by building a relationship between the electric field and magnetic field. The results showed that the uniformity of energy distribution can be improved by adjusting the radiation electric field. Sun [37] designed four kinds of structures of horn antennas (as shown in Figure 2) to enhance the uniformity of the electromagnetic field. The results showed that the electromagnetic field is evenly distributed when the length/width ratio of the antenna aperture is close to that of the feed waveguide. The homogeneity of energy distribution can be improved by adjusting the radiation electric field. Lou et al. [16] found that replacing limestone filler with ferrite filler is an effective way to improve the microwave heating uniformity of the asphalt mixture with steel slag. This is because the asphalt binders containing ferrite filler have high thermal conductivity. Fakhri et al. [38] found that adding copper slag filler into the asphalt mixtures containing steel shavings or recycled tire steel fibers can improve their temperature and heating uniformity.

In summary, researchers have acquired rich achievements in promoting the temperature uniformity of microwave heating for asphalt pavements. However, there is little research on the heat transfer mechanism in the process of microwave heating. Additionally, few researchers have carried out the simulation test from the mesoscopic perspective.

## 3. Self-Healing Properties of Asphalt Concrete under Microwave Heating

The heating technologies for accelerating the healing of the asphalt mixture mainly include induction heating, infrared heating, hot air heating, and microwave heating, as shown in Figure 3. In the process of induction heating, the heat energy diffuses into the asphalt mixture and the temperature of the asphalt pavement is increased via the Joule principle. Infrared heating is efficient, but continuous heating causes asphalt pavements to set on fire. Hot air heating has little influence on the aging of the asphalt mixture, but it will take more time. Under microwave heating, the temperature of the asphalt mixture is increased by the orientation change in polar molecules caused by the alternating magnetic field.

The self-healing behavior of asphalt concrete includes the self-healing of the asphalt binder and the self-healing of the adhesive interface between the binder and the aggregate. As early as 1939, the self-healing phenomenon of asphalt concrete was observed in the Ghrib inclined wall dam in Algeria [39]. In 1967, Bazin et al. [40] discovered the self-healing properties of the asphalt mixtures. From the macroscopic perspective, the self-healing of the asphalt mixtures is the reverse process of cracking, which can be described as crack closure and strength recovery. From the microscopic perspective, it is the spontaneous interface infiltration and molecular diffusion of asphalt molecules on the upper and lower surfaces of the cracks.

Al-Ohaly et al. [41] and Gallego et al. [42] discovered the potential of microwave heating to improve the self-healing of the asphalt mixture. Microwave heating can reduce the viscosity of the asphalt binder and accelerate its capillary flow and molecular diffusion rate by increasing the temperature, therefore improving the self-healing of the asphalt mixture. Many researchers have carried out studies on the mechanism, test methods, evaluation indexes, and influence factors of asphalt mixture self-healing.

### 3.1. Self-Healing Mechanism

The surface energy theory as well as the capillary flow theory are the major theories to explain the self-healing mechanism of asphalt concrete [43,44]. Additionally, the molecular diffusion theory and the phase field theory are commonly used to describe the self-healing principle of the asphalt mixture. The definition of surface energy is the work conducted by external substances on the object under a certain temperature and pressure. It demonstrates the self-healing principle of the asphalt mixture from the aspect of the fracture mechanics and considers that the decrease in surface energy at the crack interface promotes the healing of asphalt concrete [45,46]. The application of molecular diffusion theory can be traced back to the self-healing of polymer materials [47]. In 1981, Wool et al. [48] proposed that the crack-healing of polymer materials undergoes five stages: rearrangement, surface approach, wetting, diffusion, and molecular random distribution. Afterwards, researchers adopted the molecular diffusion theory to study the healing mechanism of asphalt concrete. Little et al. [49] proposed that the self-healing of asphalt concrete is mainly attributed to the diffusion of the asphalt binder. As the asphalt molecules at the interface diffuse to the crack, the asphalt concrete strength gradually recovers. The phase field theory uses dynamic differential equations to express diffusion, ordering potential, and thermodynamic drive. Loeber et al. [50] observed that the asphalt binder has multiphase properties. On the basis of the phase field theory, the healing of the asphalt binder can be interpreted as the process of multiphase rearrangement to form a single phase [51]. The self-healing explanation on the basis of the model of capillary flow theory assumes that the flow of the asphalt binder in the micro cracks under the driving of capillary action contributes to the closure of the cracks. Álvaro García et al. [52,53] proposed a semi-empirical model to explain the healing of asphalt mastic and concrete according to the capillary flow theory. They considered that the capillary flow in cracks is the main cause of healing. When the binder shows as a Newtonian fluid at a high temperature and the junction of the crack makes contact, a pressure difference is generated at the contact point, which can be considered as the driving force of the capillary flow of the binder. In summary, researchers have comprehensively explained the self-healing mechanism of asphalt binders and asphalt mixture.

### 3.2. Evaluation Indicators of Self-Healing Properties

For evaluating the self-healing capacity of the asphalt binder and asphalt concrete, many researchers have conducted the three-point bending test [5], semi-circular bending test [54], DSR test [55], and ultrasonic technology [56] by the way of failure, healing, and failure. The self-healing indicators are mainly calculated based on dissipated energy, fatigue life, and dynamic shear modulus. Lou et al. [57] proposed the healing efficiency level (*HEL*) to evaluate the self-healing properties of asphalt concrete, as shown in Equation (4).
(4)$$HLE\left(\%\right)=\frac{S_{e}}{S_{w}}\times 100\%$$
where $S_{e}$ is the effective healing area of the sample in cm^2^; $S_{w}$ is the whole surface area of the sample. The effective healing area is defined as the area where the surface temperature threshold is between the softening point of the asphalt binder and 90 °C.

Wang et al. [55] defined the healing index (*HI*, %) as in Equation (5) to estimate the healing potential of asphalt concrete.
(5)$$HI=\frac{G_{2}-G_{1}}{G_{0}-G_{1}}$$
where $G_{0}$ refers to the initial $G^{*}$ value, and $G_{1}$ and $G_{2}$ represent the complex shear modulus before and after the rest, respectively.

Liu et al. [58] proposed the fatigue life extension ratio to evaluate the healing performance of asphalt concrete. It is defined as the ratio of the obtained extra fatigue life after damage due to healing and due to its original fatigue life. Li et al. [25] defined the *HI* from the perspective of dissipated energy, as shown in Equation (6).
(6)$$HI=\frac{W_{Second}}{W_{First}}=\frac{\sum_{i=1}^{n_{Second}}DE_{i}}{\sum_{i=1}^{n_{First}}DE_{i}}=\frac{\sum_{i=1}^{n_{Second}}\pi \varepsilon_{i}^{2}G_{i}^{*}sin\delta_{i}}{\sum_{i=1}^{n_{First}}\pi \varepsilon_{i}^{2}G_{i}^{*}sin\delta_{i}}$$
where $W_{First}$ and $W_{Second}$ are the accumulative dissipated energy of the first and second fatigue loading, respectively, $DE_{i}$ is the dissipated energy of the *i*-th loading cycle, and $\varepsilon_{i}$, $\delta_{i}$, and $G_{i}^{*}$ are the shear strain, phase angle, and complex modulus of the *i*-th loading cycle, respectively.

Shan et al. defined the HI by the ratio of the area between the healing curve and the initial curve to the area below the initial curve [59]. It can be calculated by the following equation.
(7)$$HI=\frac{A_{d}}{A_{before}}$$
where $A_{d}$ is the area between the healing curve and the initial curve, and $A_{before}$ is the area below the initial curve.

In summary, the self-healing performance of the asphalt mixtures can be quantitatively evaluated by the above indicators. However, it is difficult to compare the results between different studies because there is no relevant standard for the self-healing test and evaluation of the asphalt mixtures. In addition, the self-healing and modulus recovery after rest are different. The existing evaluation indicators are unable to discriminate them by tests because the asphalt mixture is a complex composite material, which needs to be solved in future research.

### 3.3. Factors Influencing Self-Healing Properties

The influence factors of the self-healing performance of asphalt concrete under microwave heating can be categorized into external factors and internal factors. The external factors mainly refer to temperature, microwave heating mode, and time. The internal factors mainly involve material properties [60,61,62]. Wang et al. [28] and Fakhri et al. [38] studied the influence of microwave heating mode on asphalt concrete healing. The results indicated that intermittent microwave heating has a better healing effect compared with continuous heating. This is because the effective healing time is longer under intermittent heating. Norambuena-Contreras et al. [63] studied the self-healing properties of fiber-reinforced asphalt concrete under different microwave heating time using the three-point bending test and observed the crack size before and after healing. The results, shown in Figure 4 and Figure 5, showed that the longer the heating time, the better the healing level. Forty seconds is the optimum heating time among the considered cases. The cracks can close under a long microwave heating time, which explains the cause of the self-healing of asphalt concrete.

The material type also has an effect on the healing performance. Gonzalez et al. [64] studied the effect of RAP and metal fibers on the crack-healing of asphalt concrete under microwave heating. The results showed that the healing level is enhanced by adding metal fibers, but the opposite results occurred when adding RAP. In addition, asphalt concrete containing metal fibers and up to 30% RAP has the potential to repair cracks under microwave heating. Zhu et al. [65] evaluated the fracture-healing properties of AC-13 mixtures with 70# asphalt binder and SBS-modified asphalt binder under microwave heating via the SCB test. It was found that the SBS asphalt mixture has better healing performance at high temperatures (over 80 °C).

This can be interpreted as the special healing mechanisms of AC-13 mixture with SBS-modified asphalt, that is, the combined effect of the flow diffusion of the asphalt binder and the elastic recovery of SBS segments. Deng et al. [15] developed an asphalt mixture incorporating manganese dioxide powder (MDP) and steel fiber (SF), and investigated its heating–healing capacity through comparing with the mixtures only with MDP or SF. The results showed that the developed mixture exhibits better microwave healing performance. Wang et al. [66] found that the asphalt mixture containing carbon fiber has good healing performance under microwave heating. Two percent IM8-modified asphalt mixture has a better healing effect compared with 4% IM8-modified asphalt mixture. The reason for this is that the high content of carbon fiber may limit the Newtonian flow of the asphalt binder in the healing process. Phan et al. [67] analyzed the applicability of steel slag to promote the self-healing of the asphalt mixtures under microwave heating. They found that asphalt mixtures with 30% steel slag aggregate and 2% steel wool fibers have good healing effectiveness. Lou et al. [16] studied the self-healing performance of steel slag asphalt concrete containing ferrite fillers under microwave heating. It was found that an appropriate addition of ferrite can improve the average temperature and healing ratio of asphalt concrete. The optimal proportion of replacing limestone fillers with ferrite is 20% by volume among the considered cases. Adding excessive ferrite has a negative effect because abundant ferrite may lead to the agglomeration issue and obstruct the heat transfer process. The hardening of the asphalt binder also has adverse effects on the flow and diffusion process. Lou et al. [57] replaced the 4.75–9.5 mm aggregate of the asphalt mixture with hot braised steel slag (HBSS) in a certain volume percentage and studied its microwave self-healing performance via the SCB test. The results showed that the addition of HBSS promotes the healing of the asphalt mixture. However, excessive HBSS causes an overheating problem and reduces the effective healing area under microwave heating. González et al. [68] studied the crack-healing capability of normal asphalt mixtures and asphalt mixtures containing steel fiber, metal shavings, and silicon carbide via three-point bending tests. It was found that the additives (i.e., steel fibers, metal shavings, and silicon carbide) have little effect on crack-healing ability. The asphalt mixture without additives also showed good crack-healing ability. This is because the aggregates contain metals, as shown in Figure 6. In summary, asphalt binder is a temperature-sensitive and viscoelastic material. MSAs have good thermal conductivity performance, which enhances the diffusion rate of asphalt molecules and improves the healing ability of the asphalt mixture.

Some researchers also introduced microcapsules containing rejuvenator into asphalt concrete to promote self-healing. When the crack extends to the position of a microcapsule, it breaks and releases the rejuvenator, which accelerates the healing of asphalt concrete. Wan et al. [69] synthesized calcium alginate/nano-Fe_3_O_4_ composite capsules that can actively and rapidly release the rejuvenator under microwave heating. Kargari et al. [54] studied the self-healing effect of the asphalt mixture containing palm oil capsules under microwave heating. They found that the HI values of aged and non-aged asphalt mixtures heated by microwaves increase by 32% and 7%, respectively, when the added palm oil capsule accounts for 0.7% of the asphalt mixtures by mass. The reason for this is that microwave heating can control the release of rejuvenators in capsules. Thus, the released rejuvenator improves the healing effect of the asphalt mixtures.

In summary, most researchers explain the self-healing phenomenon of the asphalt mixture using macroscopic experimental study, but pay little attention to the influence mechanism from a multiscale perspective.

## 4. Asphalt Pavement De-icing Using Microwave Heating

Icy roads increase traffic accidents and pavement maintenance time and cost, reduce driving speed [70,71,72,73,74], and raise fuel consumption and exhaust emissions [75]. Pavement de-icing is significant to ensure traffic safety and efficiency. Traditional pavement de-icing methods mainly include the manual method, mechanical method [76], and chemical method [77]. The manual method is inefficient and costly. The mechanical method may cause damage to the pavement [78]. The chemical method usually damages vegetation, water sources, and atmosphere [79], and reduces pavement durability [80]. For overcoming the disadvantages of traditional methods, researchers have explored the applications of microwave heating [81], induction heating [82], Joule heating [83], solar energy [84], and environmentally friendly de-icing agent [85] in asphalt pavement de-icing. Microwave de-icing, developed in recent years, has great potential and application prospects. Figure 7 depicts the working mechanism of de-icing using a microwave heating vehicle.

### 4.1. Microwave De-icing Mechanism and Efficiency Evaluation

The freezing adhesion between the ice and the pavement is the major resistance of de-icing. The adhesion strength can be defined as in Equations (8) and (9). The conditions of producing freezing adhesion mainly include the temperature of 0 °C or below, certain water content, and freezing time. Temperature is the most important factor. The higher the temperature, the lower the horizontal adhesion strength of the interface between ice and pavement [87] and the crushing strength of ice [88]. Because ice has a low dielectric constant and poor microwave-absorbing ability, it is difficult to heat directly using microwaves [89]. However, the microwave can penetrate the ice layer and be absorbed by pavement materials. When the pavement temperature rises, the heat is transferred to the ice layer, which reduces the adhesion strength between the ice and the pavement [90]. Thus, it is easy to use mechanical devices to break and clean the ice layer [91].
(8)$$c_{\sigma }=\sigma =F_{1}/S$$
(9)$$c_{q}=\tau =F_{2}/S$$
where $c_{\sigma }$ is the normal freezing adhesion coefficient, $c_{q}$ is the tangential freezing adhesion coefficient, $F_{1}$ is the normal pull-out force, $F_{2}$ is the tangential pull-out force, and S is the area of the frozen interface.

Some indicators were developed for quantifying microwave de-icing efficiency. Tang et al. [92] proposed that the surface power density of the asphalt mixture can be used to evaluate de-icing efficiency. Jiao et al. [93] pointed out that the time for the ice–pavement interface temperature to reach 0 °C can be used as an evaluation indicator of microwave de-icing efficiency. Li et al. [94] found that the ice–pavement adhesion fails when the temperature reaches 3 °C. They believed that the time for the ice–pavement interface temperature to reach 3 °C is more accurate to evaluate de-icing efficiency. Gao et al. [95] evaluated the de-icing efficiency via the time for the ice to fall off the asphalt mixture under a horizontal force. Liu et al. [19] proposed that the ice melting speed (IMS), which is defined as in Equation (10), can also reflect the de-icing efficiency to a certain extent.
(10)$$IMS=\frac{m_{1}-m_{2}}{t}$$
where *IMS* is the ice melting speed, g/s; $m_{1}$ is the mass of the sample and ice before heating, g; $m_{2}$ is the mass of the sample and ice after heating, g; and t is the heating time, s.

### 4.2. Microwave De-icing Characteristics

The characteristics of microwave de-icing involve multiple factors, such as microwave frequency, microwave electric field strength, pavement materials, ambient temperature, ice thickness, and the microwave heating method [96]. As early as 1986, Monson began the research into microwave de-icing characteristics [97]. Some additives and alternatives, such as anthracite [98] and taconite [99,100], were included into the asphalt pavement to improve the de-icing capacity. Zhao et al. [101] developed a kind of asphalt mixture that can improve de-icing efficiency, in which magnetic powder completely or partially replaces limestone powder and magnetic metallurgical slag partially replaces natural coarse aggregate. Zhao et al. [102] also prepared a kind of asphalt concrete suitable for microwave absorption by completely or partially replacing calcareous mineral powder with magnetic powder and partially replacing natural aggregate with magnetic sand or silicon carbide sand. It is found that the heating rate of the new asphalt concrete is several times that of traditional asphalt concrete. Wang et al. [103] proposed a preparation method of microwave-absorbing asphalt mixture by adding hydroxy iron powder of 3~7% for improving de-icing efficiency. The results showed that the surface temperature of the material rises rapidly after microwave heating. Wang et al. [104] selected hydroxy iron power, ferroferric oxide, alumina, and expanded graphite as microwave-sensitive coating materials and carried out an ice melting test. It was found that ferroferric oxide and expanded graphite have better microwave-absorbing and ice-melting abilities. Jiao et al. [96] studied the effects of ambient temperature on microwave de-icing efficiency. They found that the lower the environmental temperature, the lower the de-icing efficiency. Some researchers focused on an improvement in microwave heating facilities. Guan et al. [105] designed a schematic of a microwave de-icer. Yang et al. [106] designed a horn antenna for a microwave de-icing device to improve heating uniformity and analyzed its heating characteristics and voltage standing wave ratio. The results showed that the microwave de-icing effect is the best when the slope angle of the horn antenna is 15°. Tang et al. [92], Jiao et al. [93], and Ding et al. [107] investigated the effects of microwave frequency on de-icing efficiency via simulation and laboratory tests. The results showed that the de-icing efficiency of 5.8 GHz microwaves is more than four times that of 2.45 GHz microwaves. The reason for this is that the microwave with a frequency of 5.8 GHz has a shallower penetration depth and higher power density, which can quickly increase the pavement temperature and improve the de-icing efficiency. Table 3 lists the de-icing efficiency of the asphalt mixtures containing different MSAs using 2.45 GHz microwaves. As can be seen from the table, the higher the environmental temperature (ET), the greater the improvement in the de-icing efficiency. The thicker the ice thickness (IT), the lower the de-icing efficiency.

In summary, most investigations focus on the addition of MSAs for improving de-icing efficiency, while there are fewer studies on microwave frequency.

## 5. Asphalt Aging in Microwave Heating Process

Both the applications of microwave for self-healing and de-icing are based on heating the asphalt pavement, which may lead to asphalt pavement aging. It is necessary to study the asphalt aging caused by microwave heating. Flores et al. [109] compared asphalt aging induced by microwave heating and infrared radiant heating through the tests of penetration, softening point, and rheological properties. The results showed that microwave heating has less of an effect on the asphalt binder. Wu et al. [110] found that microwave heating has no perceptible negative effect on asphalt aging compared with TFOT by analyzing the indicators of penetration, ductility, and softening point. Fernandez et al. [111] quantified the effects of microwave heating and long-term aging on the rheological and chemical performance of recovered asphalt binders using a frequency sweep test and Fourier transform infrared spectrometry (FTIR) analysis. The results showed that microwave heating has little influence on the aging of the rheological properties of asphalt binders. With the increase in microwave heating and long-term aging cycles, the carbonyl and sulfoxide indices increase in both phases. Lou and Sha et al. [112,113] evaluated the physical and rheological properties and infrared spectra of asphalt binders obtained from steel slag asphalt mixtures under different microwave heating cycles via the tests of penetration, softening point, DSR, and FTIR spectroscopy. The results indicated that the penetration value as well as the softening point get worse during the first 10 microwave heating cycles. Compared with the aging methods of RTFOT and PAV, microwave heating has no significant effect on the deterioration of the physical properties of asphalt binders. In summary, several researchers have pointed out that microwave heating has little effect on asphalt aging. However, it is necessary to completely evaluate the aging of asphalt pavements under cycling microwave heating.

## 6. Summary

This paper reviewed asphalt pavement self-healing and de-icing using microwave heating. Some findings are summarized as follows.

(1) Microwave sensitivity and heating uniformity are two important aspects of microwave heating properties for asphalt pavements. They determine the microwave absorption ability and heating efficiency of the asphalt mixtures and have an important effect on the self-healing quality and de-icing efficiency. Adding MSAs (such as graphite powder, magnetite powder, steel slag aggregates, and steel fiber) into asphalt mixtures can improve microwave sensitivity and heating uniformity.

(2) Microwave heating promotes the self-healing of asphalt pavements by reducing the viscosity of asphalt binders and accelerating the capillary flow and molecular diffusion rate. Several indicators have been established for quantitatively evaluating the self-healing performance of asphalt mixtures, but the results between different studies are incomparable. The reason for this is that there is no corresponding standard for the self-healing test and assessment of the asphalt mixtures. In addition, most indicators are based on performance tests rather than the self-healing mechanism of asphalt pavements.

(3) Microwave heating mode and time as well as material properties have an important influence on the self-healing performance of asphalt pavements. The healing level of asphalt pavements gradually increases with the increase in the microwave heating time. The addition of MSAs also contributes to self-healing. This is mainly because MSAs can improve microwave-absorbing ability. However, the effect of MSAs on pavement performance should be studied in depth.

(4) Pavement materials and microwave frequency are two important factors affecting de-icing efficiency. Most investigations focus on the addition of MSAs for improving de-icing efficiency, while there are fewer studies on microwave frequency. The influence of microwave frequency should be further studied. In addition, many studies on de-icing characteristics are based on laboratory simulation experiments. Field tests should be conducted to verify the indoor test results.

## 7. Outlooks

Extensive research has been conducted in the past to study the application of microwave heating to asphalt pavement self-healing and de-icing. The following recommendations for future studies are summarized.

(1) The cost of MSAs and their effect on pavement performance are often ignored in studies. It is suggested to study the feasibility of MSA application in actual projects.

(2) There is no corresponding standard for the self-healing test and assessment of the asphalt mixtures. It is necessary to develop a unified test and assessment standard of asphalt mixture self-healing.

(3) Microwave heating technology has good prospects in promoting asphalt pavement self-healing and de-icing. However, microwave heating can also result in asphalt aging. Although several researchers have pointed out that microwave heating has little effect on asphalt aging, it is also necessary to completely evaluate the aging of asphalt pavements under cycling microwave heating.

## Figures and Tables

**Figure 1 polymers-15-01696-f001:**
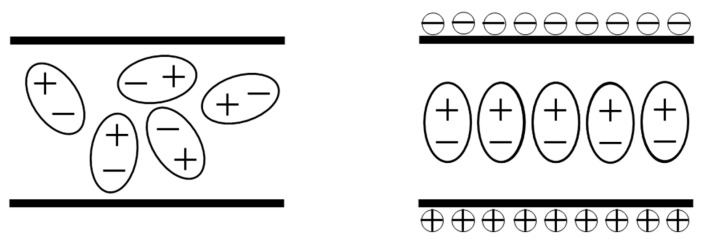
The microwave-absorbing principle of materials.

**Figure 2 polymers-15-01696-f002:**
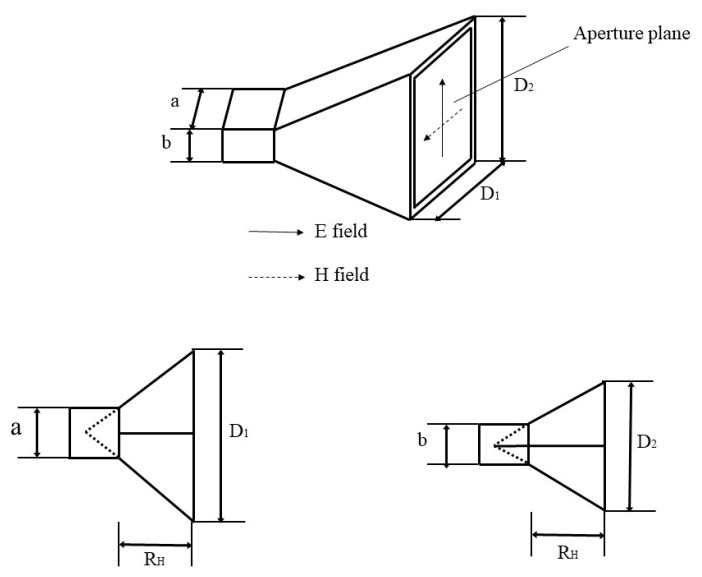
Structure of horn antenna [37].

**Figure 3 polymers-15-01696-f003:**
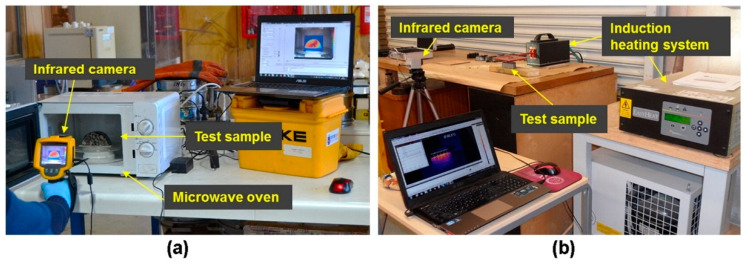
Heating devices: (**a**) microwave heating; (**b**) induction heating [7].

**Figure 4 polymers-15-01696-f004:**
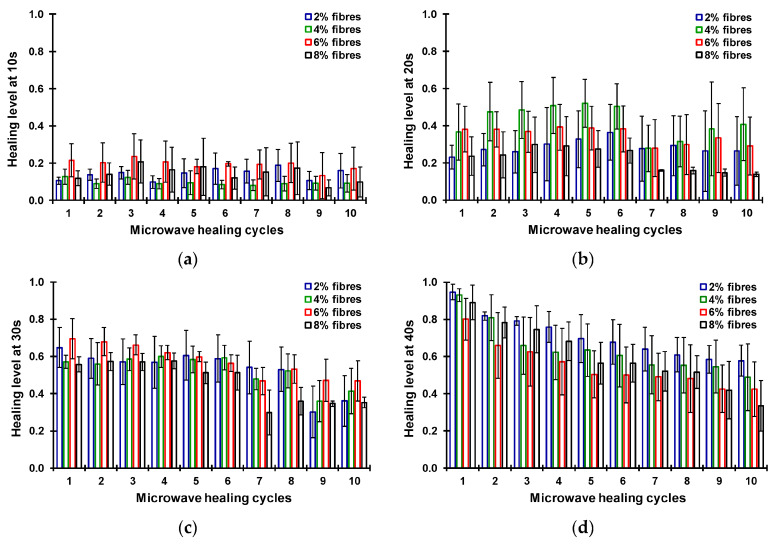
Healing levels of the asphalt mixtures under different microwave heating times: (**a**) 10 s; (**b**) 20 s; (**c**) 30 s; (**d**) 40 s. [63].

**Figure 5 polymers-15-01696-f005:**
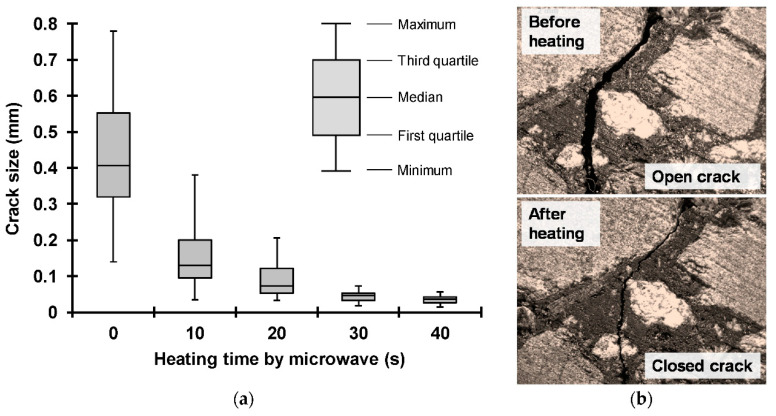
The crack size of the asphalt mixture samples: (**a**) Box plot representation of the crack size; (**b**) Images of a cracked specimen [63].

**Figure 6 polymers-15-01696-f006:**
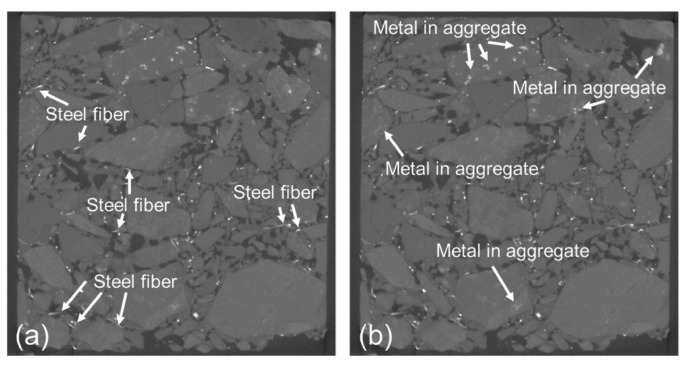
CT scan image of the asphalt mixture: (**a**) Steel fiber; (**b**) Mental contained in aggregate [68].

**Figure 7 polymers-15-01696-f007:**
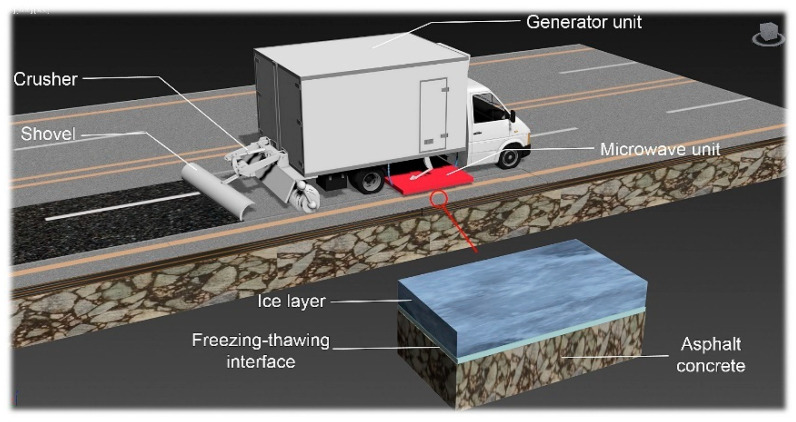
Schematic diagram of de-icing using microwave heating vehicle [86].

**Table 1 polymers-15-01696-t001:** The reflection loss values of representative materials.

Materials	Electromagnetic Frequency (GHz)	Thickness (mm)	RL (db)
Asphalt carbon-coated graphene/magnetic NiFe_2_O_4_-modified multi-wall carbon nanotube composites [9]	4.6	3.2	−45.9
Asphalt carbon-coated reduced graphene oxide/magnetic CoFe_2_O_4_ hollow-particle-modified multi-wall carbon nanotube composites [10]	11.6	1.6	−46.8
Mg-Al layered double hydroxides (LDHs) [11]	15.71	8	−4.79
LDHs:Fe_3_O_4_ = 1:1	11.71	10	−6.88
LDHs:Fe_3_O_4_ = 2:1	11.88	10	−5.25
LDHs:Fe_3_O_4_ = 1:2	11.28	8	−10.73
SiC [12]	16.12	28	−12.53
SiC:Fe_3_O_4_ = 1:1	11.21	30	−18.93
SiC:Fe_3_O_4_ = 2:1	11.26	29.5	−15.82
SiC:Fe_3_O_4_ = 1:2	17.92	26	−22.18
LDHs [13]	15.49	8	−5.21
SiC attached LDHs	17.5	10	−13.65
SiC:Fe_3_O_4_ is 3:1 [14]	2.45	25	−28
Limestone filler [15]	16.1	-	−6.8
Manganese dioxide powder	11.6	-	−18.83
Carbon powder	2.36	-	−33.53
Ferrite powder	12.3	-	−41.68
Limestone [16]	15.88	-	−2.67
Ferrite	3.89	-	−10.62
Ferrite	13.67	-	−30.28
Fine SiC [17]	13.68	2	−22.34
Fine SiC	2.45	10	−13.55
Coarse SiC	8.08	10	−15.27
Coarse SiC	2.45	10	−10.51
Asphalt mixture added with natural magnetite power in grade of 80 [18]	2.9	30	−38
Asphalt mixture added with natural magnetite power in grade of 70	3	30	−27
Asphalt mixture added with natural magnetite power in grade of 60	3.15	30	−25

**Table 2 polymers-15-01696-t002:** The heating rate of different asphalt mixtures.

Asphalt Mixtures	Aggregates	Heating Rate (°C/s)
Normal asphalt mixture [27,28,29,30]	Basalt	0.252–0.76
Normal asphalt mixture [14,17,31]	Limestone	0.2–0.548
Normal asphalt mixture [31]	Dolomite limestone	0.618
Normal asphalt mixture [31]	Granite	0.757
Normal asphalt mixture [6]	Andesite	0.355
Asphalt mixture with steel slag aggregates [27]	Basalt	0.623
Asphalt mixture with SiC and Fe_3_O_4_ powder [14]	Limestone	0.244–0.367
Asphalt mixture with SiC aggregates [17]	Limestone	0.458–0.476
Asphalt mixture with steel fiber and graphite [28]	Basalt	0.9–1.02
Asphalt mixture with graphite powder and magnetite powder [29]	Basalt	0.372
Asphalt mixture with aggregates coated by magnetic Fe_3_O_4_ films [30]	Basalt	0.888–0.9
Asphalt mixture with nano-graphite [31]	Limestone	0.579–0.815
Asphalt mixture with nano-graphite [31]	Dolomite limestone	0.658–0.92
Asphalt mixture with nano-graphite [31]	Granite	0.831–1.184
Asphalt mixture with steel fiber [6]	Andesite	0.804
Asphalt mixture with steel slag aggregate [6]	Andesite	0.696

**Table 3 polymers-15-01696-t003:** De-icing efficiency of the asphalt mixtures containing different MSAs using 2.45 GHz microwaves.

MSAs	Volume/Mass Fraction of MSAs	IT (mm)	ET (°C)	Efficiency Improvement
Ferrite [93]	10%	10	−10	3.1 times
		10	−15	2.8 times
		10	−19	2.9 times
Magnetite [21]	80%	10	−5	8.6 times
		10	−10	8.1 times
		10	−15	6.3 times
Steel slag [86]	80%	-	−5	3.1 times
		-	−20	2.6 times
MHCs [22]	100%	30	−10	5.8 times
		20	−10	6.1 times
		15	−10	6.6 times
		10	−10	8.9 times
		5	−10	9.2 times
		10	−20	7.8 times
2# steel wool fibers [95]	1%	-	−5	7 times
		-	−10	5.5 times
0# steel wool fibers	0.7%	-	−5	4.6 times
		-	−10	4.4 times
000# steel wool fibers	0.3%	-	−5	3.9 times
		-	−10	3.1 times
ACP [19]	100%	50	−15	2.5 times
Carbon fiber [108]	0.45%	-	−10	2.7 times

## Data Availability

Not applicable.

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
