# Peer review of "Review of the Application of Microwave Heating Technology in Asphalt Pavement Self-Healing and De-icing"

_polymers, 2023, doi:10.3390/polym15071696_

Round 1

Reviewer 1 Report

In this manuscript, authors reported a review paper that deals with the properties of asphalt concrete after microwave heating were summarized, including microwave sensitivity and heating uniformity. Then, the evaluation indicators and influence factors of the self-healing properties of asphalt mixtures heated by microwave were reviewed. Finally, the application of microwave heating in asphalt pavement de-icing was explored. The manuscript presents some interesting knowledge in the field of microwave heating technology in asphalt pavement for self-healing and de-icing. I reviewed the manuscript in a critical manner and some of the comments are given below:

General comments

The manuscript might be a contribution of interest for “Polymers” and in principle within its specific scope but it is not suitable for publication in this form. The manuscript lacks clear statements and critics from the authors and well stated outlook. The quality of writing is ‘more or less’ good with some grammar and spelling errors.

Moreover, the Review mostly summarizes the works in the literature rather than provide some assessments of the reported results. A good review should distill key information rather than simply compile it. Finally, you don’t have an outlook section; a good review should be foresightful and forward looking. Please incorporate a detailed outlook of the field. I look forward to see your revised version.

Recommendation: Reconsider after minor revisions noted.

Specific comments

1.      Introduction needs to be heavily improved.

2.      As I mentioned, the review only copy and summarize. It is difficult to find the authors’ deep thoughts and bullet critics about this field. In other words, there are so few comments about the published cited papers. For a better review, the negative and positive comments and suggestions are more important, which can point out the new directions from the previous publications.

3.      A review paper should be illustrative; few figures are prepared by the authors. It is expected that 4 to 8 figures should be summarized by the authors.

4.      The potential citation points are so few. The authors should give out more potential citation points for readers which could easily be done by commenting on the negative and positive sides of the mentioned methods.

5.      As I mentioned before, a review should be foresightful and forward looking therefore I suggest to incorporate an outlook section where you describe the possible scenarios for future development of self-healing and de-icing asphalt pavement.

Reviewer 2 Report

In this review article, the authors discussed the various aspects of asphalt self-healing and de-icing using microwave treatment. Through the discussion, the authors pointed out several areas that are worth further investigation:

1) the feasibility of applying MSA as a way to modulate microwave sensitivity and heating uniformity of asphalt pavements;

2) the necessity of developing a unified standard for evaluating the self-healing performance of asphalt pavements;

3) the specific mechanism through which MSA impacts the self-healing performance of asphalt pavements;

4) the effect of microwave frequency on de-icing efficiency;

5) the effect of microwave on asphalt aging.

Overall, the discussion in this review is thorough and methodical. The various aspects of microwave treatment of asphalt were discussed with good literature support and quantitative analysis. The reviewer would recommend the publication of this manuscript after addressing the following:

1) It would elevate the discussion herein by putting the topic into perspective. Some context on the potential alternative methods for inducing asphalt self-healing and de-icing could help readers to better understand the importance of studying the effect of microwave treatment on asphalt performance. Also, it would be helpful to provide a comparison between the microwave method and other potential treatments.

2) Some discussion on the current hypothesis of the effect that MSAs have on asphalt on a more fundamental level (chemical transformation or physical explanation) could help guide the discussion and provide further guidance for future study in the related area. 

Reviewer 3 Report

Please revise Fig. 1: (mental plate)?

Section 3.2: I believe there is a big difference between self-healing and modulus recovery after rest. In fact, the authors combine various perceptions of self-healing (diffusion, modulus recovery) and it is unclear which is correct for them (from their perspective), or on which you will base your analysis. This is particularly important when deciding which evaluation will be performed. A combination of physical and mechanical evaluation maybe?

Section 5: I do not believe that FTIR can show what happens to asphalt after heating: you expect the binder to lose light components or just to re- accommodate itself. 

General comment: It is necessary that the authors not only review and summarize the sources but to discuss and give their technical points of view on what they are showing. 

Reviewer 4 Report

The article is interesting,  but there are aspects to imporve in the article:

"self-healing and de-icing" must introduce in which context you are referring and in whcih situations

Which contributions presents your work?

Which effects has the Electromagnetic frequency (GHz) and RL (db) over features materials?

Enhance the main conclusions of the work

Do you believe is suitable to heat the asphalt with microwave? Whith which type of machine can be achieved this?
